# Ubiquilin Networking in Cancers

**DOI:** 10.3390/cancers12061586

**Published:** 2020-06-15

**Authors:** Salinee Jantrapirom, Luca Lo Piccolo, Dumnoensun Pruksakorn, Saranyapin Potikanond, Wutigri Nimlamool

**Affiliations:** 1Department of Pharmacology, Faculty of Medicine, Chiang Mai University, Muang, Chiang Mai 50200, Thailand; salinee.jan@cmu.ac.th (S.J.); saranyapin.p@cmu.ac.th (S.P.); 2Omics Center for Health Science, Faculty of Medicine, Chiang Mai University, Muang, Chiang Mai 50200, Thailand; lopiccolo.l@cmu.ac.th (L.L.P.); dumnoensun.p@cmu.ac.th (D.P.); 3Department of Orthopedics, Orthopedic Laboratory and Research Network Center (OLARN), Faculty of Medicine, Chiang Mai University, Chiang Mai 50200, Thailand; 4Excellence Center in Osteology Research and Training Center (ORTC), Chiang Mai University, Chiang Mai 50200, Thailand; 5Research Center of Pharmaceutical Nanotechnology, Chiang Mai University, Chiang Mai 50200, Thailand

**Keywords:** ubiquilins, UBL-UBA, proteasome, autophagy, genetic variants, cancers

## Abstract

Ubiquilins or UBQLNs, members of the ubiquitin-like and ubiquitin-associated domain (UBL-UBA) protein family, serve as adaptors to coordinate the degradation of specific substrates via both proteasome and autophagy pathways. The UBQLN substrates reveal great diversity and impact a wide range of cellular functions. For decades, researchers have been attempting to uncover a puzzle and understand the role of UBQLNs in human cancers, particularly in the modulation of oncogene’s stability and nucleotide excision repair. In this review, we summarize the UBQLNs’ genetic variants that are associated with the most common cancers and also discuss their reliability as a prognostic marker. Moreover, we provide an overview of the UBQLNs networks that are relevant to cancers in different ways, including cell cycle, apoptosis, epithelial-mesenchymal transition, DNA repairs and miRNAs. Finally, we include a future prospective on novel ubiquilin-based cancer therapies.

## 1. Introduction 

Ubiquilins (UBQLNs) are essential factors for the maintenance of proteostasis in cells since they work as adaptors to deliver poly-ubiquitinated proteins to the proteasome [1,2] and participate in autophagosome formation [3,4,5], as well as in the endoplasmic reticulum (ER)-associated protein degradation pathway (ERAD) [6].

Human genome encodes four major UBQLN proteins (UBQLN1–4) that belong to the non-proteasomal UBL-UBA family by containing an ubiquitin-like (UBL) domain at the N-terminus and an ubiquitin-associated domain (UBA) at the C-terminus (Figure 1). UBQLN1, UBQLN2 and UBQLN4 are ubiquitously expressed, while UBQLN3 is exclusively expressed in the testis. The fifth UBQLN gene, called *UBQLNL*, was later detected in humans and still very poorly characterized [7].

Given the broad physiological implication of proteostasis pathways, any dysregulation of proteostasis is often involved in the development of multiple pathological conditions. Accumulated evidence links UBQLNs to neurodegenerative diseases such as Alzheimer’s disease (AD) or another form of dementia with locomotor dysfunctions [8,9] and other proteinopathies like amyotrophic lateral sclerosis (ALS) [10,11,12]. Moreover, increasing evidence suggests a role of UBQLNs in diverse types of cancers due to their activities in the modulation of important players of cell cycle, apoptosis, membrane receptors, DNA repairs, epithelial-mesenchymal transition (EMT) and miRNAs. Nevertheless, the mechanisms of how UBQLNs are involved in tumorigenesis and cancer progression are fragmented or contradictory, making it difficult to assess the contribution of such a family of proteins to the network of human cancers.

In this review, we aim to summarize and discuss the UBQLN network that has been proved to be relevant to tumorigenesis and cancer progression.

## 2. UBQLN’s Genetic Variants in Cancers

At least six different genetic variants of *UBQLN2* have been found to clearly correlate with ALS (OMIM entry 300264 at www.omim.org). Moreover, a homozygous nonsense mutation (c.976c>T) of *UQBLN4* gene was recently discovered to cause a rare inherited disorder called ataxia-telangiectasia (A-T) [13].

To date, none of UBQLN’s genetic variants has been associated with cancers. Nonetheless, as discussed above, new studies have shown that an alteration of UBQLNs expression levels and/or a formation of proteinaceous UBQLNs-containing cytoplasmic aggregates are certain conditions that lead to abnormal cell growth and genome instability [13].

By utilizing an open platform that interactively explores multidimensional cancer genomics data sets in the context of clinical data and biologic pathways (https://www.cbioportal.org) [14,15], we were able to track several UBQLN single nucleotide variations (SNVs) and copy number variations (CNVs) that are associated with a large variety of human cancers, including breast, ovarian cancers and lung adenocarcinoma (Appendix A). For instance, we found that gene amplification is prominently found among *UBQLN* variations associated with cancers. Amplification of *UBQLN4* is associated with 14.96% of breast cancer (446 samples out of 3116 samples from both the Cancer Genome Atlas (TCGA) and non-TCGA non-redundant studies). Similarly, amplification of *UBQLN4* can also be identified in 10.12% of ovarian cancer patients (59 samples out of 583 samples from both TCGA and non-TCGA non-redundant studies) (Appendix A).

By accessing the cbioportal for *UBQLN*’s SNVs, the data have shown that some *UBQLN*’s polymorphisms segregate with the cancer samples by a significantly high allele frequency, especially in breast invasive carcinoma tumor (Appendix A). We identified *UBQLN1* non-sense mutations in 31% (Q176*) and 53% (G499*) of variants reads from the TCGA and non-TCGA non-redundant breast and lung cancers studies, respectively (Appendix A). Moreover, three missense mutations in *UBQLN2* (P440Q, H90N and G481V) and *UBQLN3* (M546I, I247L and S294L) genes, show allele frequency higher than 50% in both TCGA and non-TCGA non-redundant studies of NSCLC (Appendix A).

On the basis of these findings, it would be of interest to run new studies in order to understand the possible relevance of such *UBQLN* variants in the onset of cancers or their progression. On the other hand, it is important to explore the reliability of UBQLNs as biomarkers of human cancers.

In this regard, some studies have already tested the feasibility to include UBQLNs as new biomarker panels for clinical and prognostic purposes in diverse human cancers. For example, Shimada and co-workers showed that UBQLN2 immunostaining can be a practical test for urine cytology allowing the prediction of tumor grade and stage [16]. Profiling the UBQLN1 together with other peptides in serum samples from lung adenocarcinoma was found to be able to predict cancer status with 85% sensitivity and 86% specificity [17,18]. Moreover, clinical and prognostic significance of UBQLN1 was more deeply investigated in breast and gastric cancers as well as in NSCLC and it revealed that high levels of UBQLN1 are often associated with high histological grade, invasion and lymph node metastasis. Therefore, high UBQLN1 expression is considered to be a worse prognostic factor for patient with gastric cancer [19] and a novel marker to predict poor prognosis in breast cancer [20].

## 3. UBQLNs and p53

In order to deal with various harmful stimuli that affect genomes, our cells need a tight control of checkpoint mechanism to detect errors and suspend cell progression until the errors are certainly fixed [21,22]. Failure of checkpoint mechanism leads to cancer progression and anti-cancer drug resistance [23,24]. The tumor suppressor protein, p53 mediates various mechanisms to arrest cell cycle and allow DNA repairs or induce cell apoptosis due to an occurrence of unmanageable DNA damages [25]. With a wide spectrum of its functions participating in stress response, hypoxia, nutrient deprivation, telomere erosion, UV light, ionized radiation and chemotherapeutic reagents and so forth [26,27], it becomes an essential interactor between extrinsic factors and intrinsic biological responses.

The first documented interaction between p53 and UBQLNs has been established by Feng, et al. [28]. UBQLN which is also called “protein-linking integrin-associated protein and cytoskeleton 1” or “PLIC1” has been found to be involved in Kaposi’s sarcoma-associated herpesvirus (KSHV) replication. The virus uses its K7 membrane protein to bind to UBQLN and inhibit UBQLN-p53 interaction, thus p53 undergoes degradation. Therefore, the survival of infected cells is favored for viral replication [28].

Huang et al., has established the role of UBQLN4 – p53 in gastric cancer cell line. Overexpression of UBQLN4 induces cellular senescence and G1-S arrest via p53/p21 axis. p21 or p21^cip1/waf1^ is a vital player mediating G1-S cell cycle arrest by binding and inhibiting the cyclin dependent kinase 2 (cyclin-CDK2), cyclin-CDK1 and cyclin-CDK4,6 complexes. This protein is transcriptionally regulated by both p53-dependent and p53-independent mechanisms. Stabilization of p53 by UBQLN4 overexpression gives a great chance for p53 to transcriptionally regulate p21 [29]. In addition, acetylated p53 has been allowed to activate *CDKN1A*, a gene encoding p21 [30] (Figure 2A). The level of p21 can also be controlled by translational and posttranslational levels, especially by ubiquitination. Several E3 ubiquitin ligase complexes including SCF^Skp2^ [31], MKRN1 [32], CRL4^CDT^ [33], APC/C^CDC20^ [34] and RNF114 [29,35] promote p21 ubiquitination and degradation. UBQLN4 physically interacts with RNF114 and negatively regulates its expression leading to p21 stabilization [29] (Figure 2A). Therefore, UBQLN4 has been considered to be a tumor regulator through p53/p21 axis since then.

The UBQLNs-induced p53 stabilization may also explain how breast cancer cell line elicits a resistance to ionized radiation (IR) and autophagy might play an important role. In cancer cells, autophagy normally acts as a tumor suppressor by promoting non-apoptotic form of programmed cell death but it may also work as pro-survival and resistant mechanisms against stressful environments and chemo/radiotherapy [36]. An increase in autophagic function upon irradiation can prevent cancer cells from damages and vice versa [37]. Normally, UBQLNs mainly work in coordination with both UPS and autophagy [3,4,38,39,40] and the reduction of UBQLN1 affects autophagosome formation [4,5]. Thus, it is not surprising to discover another role of UBQLNs in autophagy-related mechanisms in cancers. Indeed, knockdown of UBQLN1 enhances radio-sensitivity and breast cancer cell apoptosis by suppressing irradiation-induced autophagy and when UBQLN1 is re-introduced into the cells, the radio-sensitivity is eventually reversed [41]. In this case, an autophagic suppression under UBQLN1 depletion might act through p53.

Several mechanisms are involved in radiotherapy-induced autophagy. Among them, a mechanism triggered by p53 transcriptional activation leads to induction of several factors such as the damaged regulated autophagy modulator (DRAM), Unc-51 like autophagy activating kinase 1/2 (ULK1/2), BCL-2, adenovirus E1B 19-kDa-interacting protein 3 (bnip3) and sestrin1/2 [42] (Figure 2B). In particular, Sestrin1/2 then activates TSC1/2 complex and also AMP-activated protein kinase (AMPK) to promote the autophagy induction via mTOR1 inactivation [43]. Thus, the reduction of p53 may lead to autophagic suppression rendering cells to be susceptible to IR (Figure 2B). However, a decline in the expression of the anti-apoptotic BCL2 (as discussed in the next section) is another possibility to explain an increase in radio-sensitivity caused by UBQLN1 depletion.

## 4. UBQLNs and BCL2 Family Proteins

BQLN1 has been found to specifically interact and stabilize BCLb/BCL2L10, one of the six members of the anti-apoptotic BCL2 family, via the first two of STI domains in the middle of UBQLN1 [44]. Normally, BCLb/BCL2L10 is localized in the mitochondrial and ER membranes and it is degraded in the cytoplasm by UPS [45,46,47]. Upon stimulation of UBQLN1 expression, BCLb/BCL2L10 is relocated into the cytoplasm, ubiquitinated at multiple lysine residues and stabilized, increasing a chance of tumorigenesis and progression [46]. Despite a controversial role of BCLb/BCL2L10 in apoptotic-related functions, BCLb/BCL2L10 protein expression is inversely correlated with survival outcome of patients in various cancer types [46,48,49]. BCLb binds to Bax and suppresses apoptosis induced by Bax overexpression [50]. However, further studies to define its interactive partners in each of cancer-specific condition are required. For example, the ubiquitinated and non-ubiquitinated forms of BCLb/BCL2L10 can be stabilized by UBQLN1 and BCLb/BCL2L10-UBQLN1 interaction leads to an increase in oncogenicity in leukemic models [46] (Figure 3). Consistently, patients with primary lung adenocarcinomas are reported to have highly expressed UBQLN1 mRNA and protein in their tumors, compared to an adjacent lung tissue [17].

In addition, the binding of BCLb/BCL2L10 and Beclin 1 (BECN1), a central regulator of autophagy, leads to an inhibition of autophagosome formation by disrupting BECN1-VPS34 complexes. Therefore, low expression of BCLb/BCL2L10 in hepatic cell carcinoma (HCC) tissues and cells can release free BECN1 to regulate autophagosome formation and lead to cancer cell survival [51,52] (Figure 3).

Under hypoxic condition, UBQLN2 may indirectly interact with other anti-apoptotic BCL2 family proteins, like BCL2. UBQLN2 acts as a protective molecule against cytotoxicity in osteosarcoma cell line independently to HIF-1a induction. Both JNK and p38 pathways have been found to be activated under hypoxia following UBQLN2 siRNA transfection. Such activation is known to induce apoptosis. For instance, the active phosphorylated-JNK (p-JNK) antagonizes BCL2 and induces a release of cytochrome C (Cyt C) from the mitochondrial intermembrane space through a Bid/Bax-dependent mechanism [53,54].

## 5. UBQLNs and Membrane Receptor Proteins

Substantial evidence reveals the role of insulin-like growth factor 1 receptor (IGF1R), a tyrosine kinase receptor which transmits signals to support cell proliferation, differentiation, survival and metabolism in most types of cancer [55]. Although some SNPs have been identified in IGF1R, the high expression of IGF1R and its mRNA without any mutations has been frequently discovered in transformed cells [56,57]. By looking at its functions exclusively in cancers, IGF1R activation works coordinately with several downstream pathways including PI3K/AKT, Ras/Raf/MAPK and STAT to induce oncogenic transformation, cellular outgrowth, apoptotic resistance, migration, metastasis and angiogenesis [58]. At least three ligands have been reported to bind and stimulate IGF1R. Those ligands include IGF-1, IGF-2 and insulin [59]. Once IGF1R is phosphorylated, its binding partners like IRSs and Grb2, Grb10 and Shc are recruited to further regulate downstream signals [60,61,62]. Then, if the signals are overwhelmed, a normal dephosphorylation by phosphatases takes place in order to modulate and prevent a deleterious event caused by an over-stimulation [63].

A reduction but high ratio of active/total IGF1R receptors has been found in UBQLN1-deficient human non-small cell lung adenocarcinoma, A549 cell lines. Moreover, this phenomenon has also been seen in the case of insulin receptor (INSR) and insulin-like growth factor 2 receptor (IGF2R) [64]. Kurlawala et al. reported that IGF1R and UBQLN1 physically interact via the STI-1 and STI-2 domains of UBQLN1 and this binding is crucial for the IGF1R stabilization [44]. The authors also mentioned that the mechanism of UBQLN1-dependent IGF-1 stabilization is neither through internalization nor endocytosis, because the interaction is found in several forms of IGFR including pro-IGFR, phosphorylated IGFR and normal IGFR [64] (Figure 4). However, the possible role of UBQLN1 in recruiting phosphatases to the receptors should also be considered, even though there are no current data supporting this hypothesis.

The role of UBQLNs in cell membrane receptor stabilization is not limited to IGF1R and its relatives. The regulation is likely observed in gamma-aminobutyric acid A (GABA_A_) receptor, G-protein coupled receptors (GPCRs) and nicotinic acetylcholine receptors (nAchRs) [65,66,67,68]. Moreover, B1a B-cells and BJAB B-cell lymphoma in the absence of UBQLN1 undergo cell proliferative defects after B cell receptor (BCR) ligation [69]. The defects in cell cycle entry of both kinds of B-cells are associated with hydrophobic and mitochondrial protein accumulations which can later depress cytosolic protein synthesis [70,71,72]. Indeed, the authors noted that UBQLN1 could critically be involved in the mRNA translational step of some specific cyclins required for G1 cell cycle progression [69].

## 6. UBQLNs and EMT Markers

Epithelial-mesenchymal transition (EMT) is an underlying process in cancer metastasis when epithelial cells lose their cell polarity and cell-cell adhesion and gain migratory and invasive properties [73,74]. Therefore, EMT is believed to play important roles in cancer metastasis, aggressiveness and radio-/chemo-resistance.

Among several pathways that induce EMT, a downregulation of epithelial protein, E-cadherin and a gain of the mesenchymal protein Vimentin have been found to play a critical role in cancer progression and metastasis [75,76]. Therefore, the tight regulation of these two functional opposite proteins is mandatory for cancer cells to control their aggressiveness and treatment responsiveness. Studies have shown that the expression levels of E-cadherin can be determined by the activity of various transcriptional factors such as Snail1, Snail2 (Slug), Twists, TCF3 (transcription factor 3), KLF8 (Kruppel-like factor 8) and zinc-finger E-box binding protein (ZEB1 and ZEB2) [77,78,79,80,81,82].

ZEB1, a crucial EMT activator, represses E-cadherin and activates Vimentin expression by several mechanisms. ZEB1 directly binds to E2 box type element of E-cadherin promotor [83]. It also forms a complex with Sirt-1 (a class III histone deacetylase (HDAC)) in order to bind to the E-cadherin promoter in pancreatic and prostate cancer cells [84]. In prostate cancer cells, ZEB1 plays dual functions in order to repress E-cadherin and at the same time activates Vimentin by recruiting SET8 (a histone H4K20-specific methyl transferase) to E-cadherin promoter and activate a ZEB1-dependent H4K20 monomethylation, respectively [85]. ZEB1 and UBQLN1 have been found to regulate each other in order to induce EMT in cancer cells. Although there is no direct interaction between UBQLNs and ZEB1, loss of UBQLN1 and UBQLN2 in human A549 and H358 non-small cell lung adenocarcinoma cell lines dramatically increases ZEB1 expression to repress E-cadherin and initiate cell migration and invasion. However, without ZEB1, loss of UBQLNs is insufficient to repress E-cadherin [86]. These results suggest that ZEB1 is an essential mediator of UBQLNs to regulate EMT during cancer metastasis (Figure 5).

## 7. UBQLNs and DNA Repair Proteins 

DNA double strand breaks (DSBs) is one of deleterious events of genome. It can be generated by endogenous factors such as reactive oxygen species (ROS) and exogenous factors such as ionized radiation (IR) and chemotherapeutic drugs [87,88]. DSBs can be recognized by two major DNA damaged response (DDR) pathways. The first is “Non-homologous end-joining or NHEJ”, which is fast but non-specific ligation and sometimes might lead to deletions or insertions in genomes. This pathway is generally considered to be error-prone. The second is “Homologous recombination-mediated DNA repairs or HRR” which is accurate but slow and is strictly active only in S and G2 phases whereas a homologous template is available [89,90]. The balance of these two pathways are crucial in order to achieve a correct repair. If the damages are beyond the capacity of DDR or imbalanced DDR, the affected cells will undergo death pathways or eventually tumorigenesis.

Like other two members of UBA-UBL shuttling proteins (HR23 and Ddi1), UBQLN4 has also been identified as a protein participating in DSBs repair mechanism. UBQLN4 regulates a clearance of ubiquitinated MRE11, an ATM (Ataxia Telangiectasia Mutated) substrate which functions in initiating DSB end resection [13,91]. Once DSBs occur, MRE11-RAD50-NSB1 (MRN complexes) are formed and localized into the affected area and the complexes later induce ATM autophosphorylation at Ser367, Ser1893, Ser1981 and acetylation at Lys3106 [92,93]. Several studies provide evidence that ATM, UFMylated MRE11 (a conjugation form of MRE11 by ubiquitin-fold modifier 1 (UFM1)) and MRN complexes are considered to be specific factors processing toward HRR pathway [94]. Activated ATM phosphorylates substrates which are located around chromatins to provide a scaffold for the DNA repaired complexes and control an excessive end resection by regulating a UBQLN4-MRE11 interaction (Figure 6). Phosphorylation of UBQLN4 at Ser318 facilitates MRE11 degradation through UPS and represses HRR. A mutation in UBQLN4 (c.976C>T) which is a loss of function mutation, leads the cells to manage DSBs by shifting toward HRR pathway. On the contrary, the NHEJ pathway is chosen under an UBQLN4-overexpressing condition which is found in aggressive tumors. However, these tumors are more susceptible to olaparib, poly ADP ribose polymerase (PARP) inhibitor, than be the UBQLN4-underexpressing tumors [13].

## 8. Emerging Evidence of Ubiquilins/ncRNAs Axis in Cancer

MicroRNAs (miRNAs) are small non-coding RNAs (19–22 nucleotides in length) that are known to be fundamental regulators of eukaryotic gene expression. miRNAs contribute to the regulation of a variety of biological processes, such as cell cycle, differentiation, proliferation, apoptosis, autophagy, stress tolerance, energy metabolism and immune response [95,96,97,98]. Furthermore, it is known that miRNAs play a critical role in cancer pathogenesis and the dysregulation of miRNAs is a well-known feature of cancer.

The miR-200 family is involved in the self-renewal of cancer stem cells, EMT and chemo-sensitivity [99,100,101,102]. Interestingly, in three independent online databases (Targetscan, Pictar and miRbase) UBQLN1 is identified to be the target of *miR-200c*, a member of *miR-200* family. Further in vivo experiments have proven that *mir-200c* is capable of binding to the 3’-UTR of *UBQLN1*, thus the overexpression of *mir-200c* in MDA-MB-231 and BT549 cells can lead to a strong reduction of *UBQLN1* mRNA levels [41]. Therefore, Sun and co-workers demonstrated that *miR-200c* inhibits autophagy and enhances radio-sensitivity in breast cancer cells by targeting UBQLN1 (Figure 7A).

Similarly, through online prediction Yang and co-workers observed that *miR-155* has two putative binding sites with *UBQLN1* 3’-UTR and further proved that overexpression of *miR-155* reduces the expression of *UBQLN1* mRNA and its protein in human nasopharyngeal carcinoma (NPC) cells [103]. Interestingly, *miR-155* and *antimir-155* showed abrogative and additive effects of UBQLN1, respectively; therefore, Yang and co-workers proposed that the *miR-155*/UBQLN1 axis is involved in multiple signaling pathways related to cell proliferation, cell cycle and EMT by regulating c-myc, E2F1 and cyclin D1. However, unlike the effects of *miR-200c*-dependent silencing of UBQLN1, the *miR-155*-dependent inhibition of UBQLN1 leads to an increase of radio-resistance in cancer cells because the overexpression of *miR-155* abrogates the apoptosis inducing effects of UBQLN1 [103] (Figure 7B).

In line with the Yang and co-workers findings, other reports have shown that the Diesel exhaust Particles (DEP)-dependent downregulation of *miR-155* leads to an augmentation of *UBQLN1* and *UBQLN2* mRNA levels in lung adenocarcinoma cell lines [104]. The study by Yadev and collaborators is the first to show an environmental carcinogen to regulate expression of UBQLN proteins and to suggest that UBQLNs are capable of protecting the cells from DEP toxicity. Interestingly, UBQLN1 proteins are frequently lost in lung cancer patients and the decreased UBQLN levels have proven to induce EMT in lung cancer cells [86] (Figure 7B).

The *miR-675* coding sequence lies within the first exon of the long-non-coding RNA (lncRNA) *H19* [105]. *miR-675* has been proven to cause reduction of cell proliferation, cell growth arrest and cell death and thus acts as a tumor suppressor in NSCLC and other cancer cells [106,107,108]. Interestingly, Wang and co-workers showed that *miR-675* controls the abundance of UBQLN1 in a fashion that augmentation of *miR-675* leads to an increase in the level of UBQLN1 which in turn reduces the level of ZEB1. Given the fundamental role of ZEB1 as a prime element of a network of transcription factors that controls EMT, the *miR-675*/UBQLN1 axis was shown to regulate the progression and development of pancreatic cancer by the regulation of ZEB1-dependent EMT [108] (Figure 7C).

The majority of studies, revealing the ncRNA/UBQLN axis in cancer, focus on the effects of miRNAs on UBQLN1. Recently, new findings suggested that such a fine cell regulation also involves other UBQLNs, like UBQLN4. Indeed, Yu and co-workers showed that *miR-370*/UBQLN4 axis regulates the formation and progression of hepatocellular carcinoma [109]. Given the emerging role of UBLQN4 in regulating cell proliferation and invasion [29,110], the online tools TargetScan, miRTarBase and miRcode were run to further investigate the upstream regulators of UBQLN4. In vivo studies showed that *miR-370* binds to the *UBQLN4* 3’-UTR and leads to its degradation. Interestingly, the TCGA database analysis also confirmed that the expression of UBQLN4 negatively correlates with *miR-370* expression [109]. As such, *miR-370* seems to modulate the Wnt-β-catenin pathway that is controlled by UBQLN4 and is responsible for HCC progression (Figure 7D).

## 9. Conclusions and Future Perspectives

Although increasing evidence clearly points to the crucial roles of ubiqulin proteins in many cancers, recently there has been no ubiquilin-targeted therapy available in the market or in clinical trials. Parallel drugs that targets ubiquitin-protein degradation pathway including proteasome inhibitors (bortezomib, carfilzomib and ixazomib) are currently in clinical use. However, these three drugs which are mainly US FDA approved for relapse multiple myeloma and mantle cell lymphoma, provide decent treatment outcomes with many adverse effects. For instance, bortezomib can cause permanent nerve damage to the extremities, called bortezomib-induced peripheral neuropathy (BIPN). Furthermore, cancer cells developed resistance to this drug through an enhanced aggresome-autophagy pathway, increased expression of proinflammatory macrophages, decreased ER stress response and alterations in apoptotic signaling.

Considering the structure of UBQLNs, it contains similar domains at both N- and C-terminal regions to ubiquitin and it plays important roles in trafficking ubiquitinated proteins to the proteasome. Therefore, designing certain potential molecules to suppress the function of UBQLNs is not too difficult but the major challenge is to create the agents that can specifically target the mutated forms of UBQLNs in some cancer cells so that it does not affect normal healthy cells. Additionally, since UBQLNs play diverse functions in different types of cancer, these molecules can be “good” or “bad”. Therefore, developing only for ubiquilin inhibitors may not be a perfect solution for all type of cancers. Moreover, some cancer cells exhibit proliferative characteristics and tumorigenesis due to the lack of certain ubiquilin molecules, thus it needs to consider development of effective agents that can be able to restore the production of ubiquilin in a specific manner. Lastly, therapeutic strategies should be taken into consideration to optimize the best combinatorial treatments for patients. For instance, the inhibition of ubiquilin function together with radiation may enhance radiosensitivity.

Taken together, attempting to create personalized medicine for cancer patients with ubiquilin alterations may require an additional step to measure the level of certain ubiquilin, as well as its genetic and function abnormalities before choosing the most effective treatment for the patients.

## Figures and Tables

**Figure 1 cancers-12-01586-f001:**
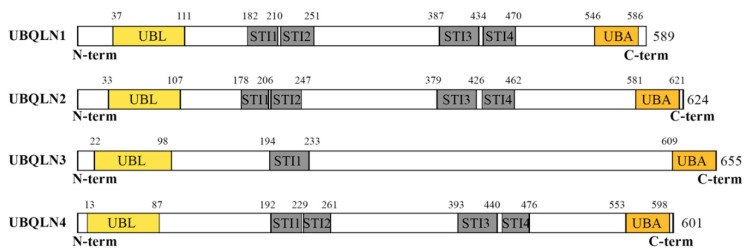
Structural organization of human ubiquilins (UBQLNs). The figure depicts the functional domains of major UBQLNs (UBQLN1-4) that have been functionally linked to human diseases. Structural analyses of the UBQLN proteins were performed using the integrated tool InterProScan (available online at https://www.ebi.ac.uk/interpro/). UBQLN1, UBQLN2 and UBQLN4 proteins have very similar UBA and UBL domains, encode structurally identical proteins containing four STI domains and are related by successive duplications [7]. In contrast, UBQLN3 carries a single STI domain and has been proposed to be more evolutionary divergent. UBL: Ubiquitin-like domain; STI: STress Inducible proteins or Hsp70–Hsp90 organizing protein; UBA: Ubiquitin associating domain.

**Figure 2 cancers-12-01586-f002:**
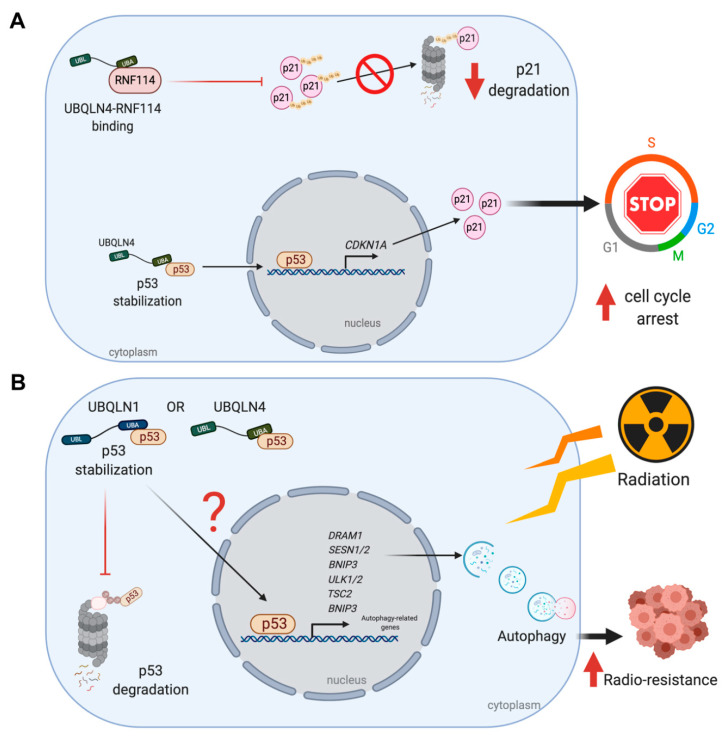
UBQLNs and p53 networking in cancers. UBQLNs stabilizes p53 and RNF114, two essential regulators of p21. (**A**) UBLQN4 physically interacts with p53, allowing the activation of p21 transcription and translation. Binding of UBQLN4 and RNF114 leads to the inhibition of p21 ubiquitination and degradation via UPS. (**B**) The stabilization of p53 by UBQLNs might induce transcription of autophagic related genes and further autophagosome formation. The autophagic induction can increase radio-resistance in cancer cells upon ionized radiation.

**Figure 3 cancers-12-01586-f003:**
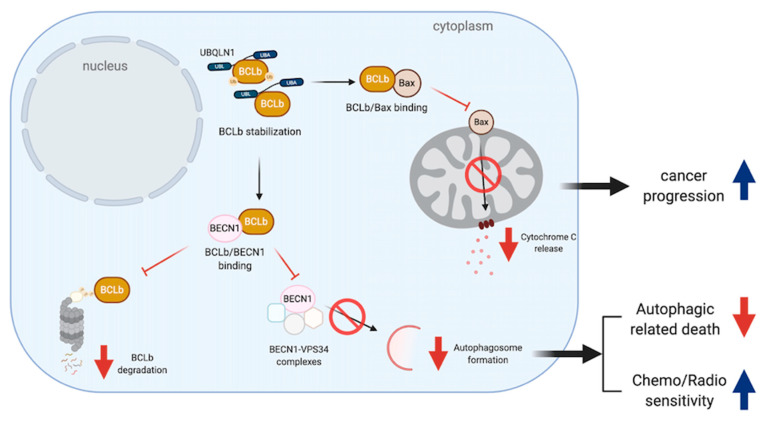
UBQLNs and BCLb/BCL2L10 networking in cancers. BCLb stabilization increases the chance of BCLb-Bax binding leading to apoptosis reduction and cancer progression. Moreover, such a stabilization can increase the chance of BCLb to bind to BECN1. BCLb-BECN1 association causes a reduction of BCLb degradation and reduces autophagosome formation by disrupting BECN1-VPS34 complexes. The reduction of autophagy affects both cell death mechanism and chemo/radio sensitivity.

**Figure 4 cancers-12-01586-f004:**
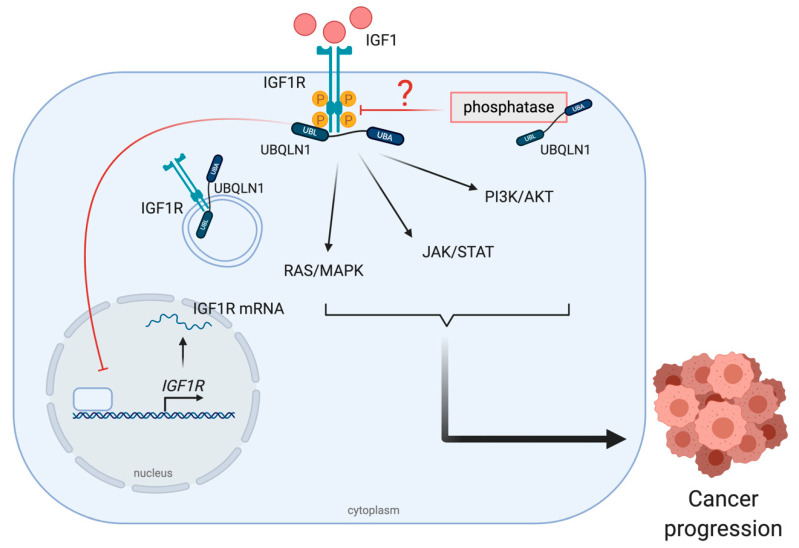
UBQLNs and insulin-like growth factor 1 receptor (IGF1R) networking in cancers. The stabilization of UBQLN1 and IGF1R can be found in various forms such as pro-form, phosphorylated form and normal form. After IGF1R activation, downstream signal transduction pathways including RAS/MAPK, JAK/STAT and PI3K/AKT, are activated to drive cancer progression.

**Figure 5 cancers-12-01586-f005:**
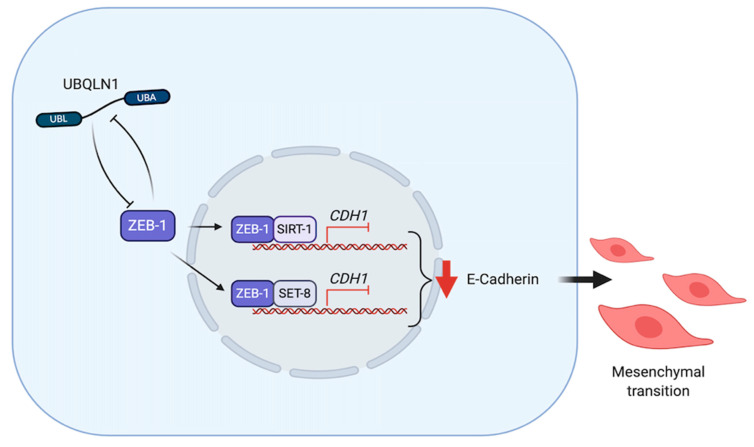
UBQLNs and ZEB1 networking in cancers. UBQLN1 and ZEB1 negatively regulate each other. UBQLN1 can repress E-cadherin expression via ZEB-1 dependent mechanism. ZEB1 forms complexes with SIRT-1 and SET-8 in order to repress E-cadherin expression and drive epithelial-mesenchymal transition (EMT).

**Figure 6 cancers-12-01586-f006:**
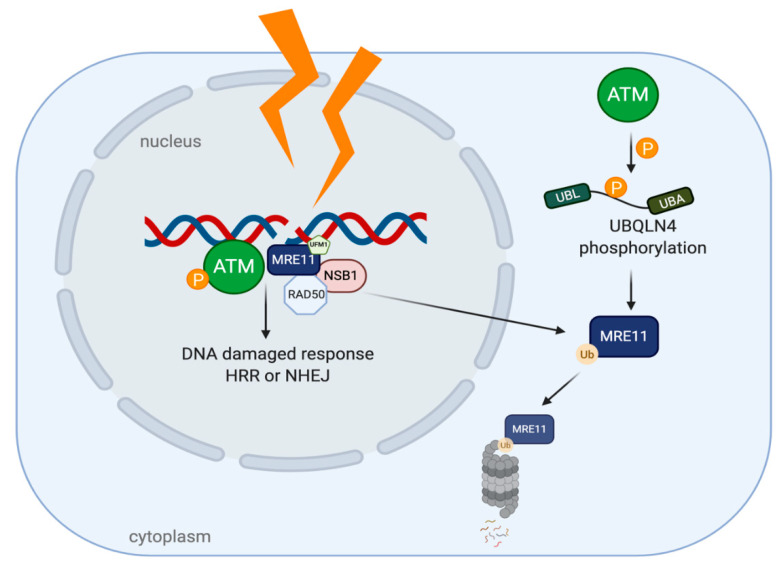
UBQLN4 and DNA repair networking in cancer. After DNA damages, Ataxia Telangiectasia Mutated (ATM) is auto-phosphorylated and MRN complexes are recruited to the damaged sites, leading to DDR. The phosphorylation of UBQLN4 by ATM can later induce MRE11 ubiquitination and degradation by UPS.

**Figure 7 cancers-12-01586-f007:**
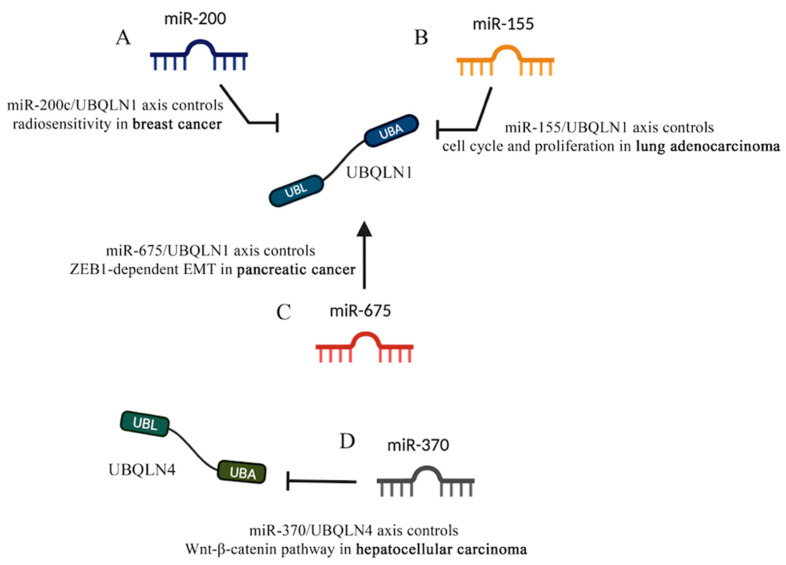
UBQLNs and miRNAs networking in cancers. Different kinds of miRNAs affect UBQLN’s functions that are relevant to cancer. (**A**) miR-200 and (**B**) miR-155 negatively control UBQLN1 functions in breast cancer and lung adenocarcinoma, respectively. (**C**) miR-675 positively controls UBQLN1 in pancreatic cancer. (**D**) The negative control of UBQLN4 by miR-370 in hepatocellular carcinoma is shown.

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
