# Peer review of "Ubiquilin Networking in Cancers"

_cancers, 2020, doi:10.3390/cancers12061586_

Round 1

Reviewer 1 Report

In this review, the authors summarized the pathophysiological functions of ubiquilin (UBQLN) family proteins on tumorigenesis. Characteristically, UBQLNs contain the N-terminal ubiquitin-like (UBL) and the C-terminal ubiquitin-associated (UBA) domains, and regulate multiple cellular functions. Importantly, genetic mutations in UBQLN are reportedly associated with amyotrophic lateral sclerosis (ALS). Therefore, the involvement of UBQLNs in neurodegenerative diseases have been extensively investigated hitherto. Intriguingly, this review sheds light on the role of UBQLNs on cancers by regulating prognosis, p53, cell death, membrane receptors, EMT, and miRNAs. Thus, this review will be helpful for all the readers who have interests on UBQLNs and tumorigenesis.

  1. On lines 31-35; Although the authors introduced the UBQLN1-4 and UBQLNL in this review, the existence of UBQLN5 and UBQLN6 are currently proposed in the Ref [7]. If it is possible, I hope the authors to represent the total residues and domain structures of human UBQLN family proteins as a figure, since the authors refer to the STI domains (on lines 148 and 198), whereas only UBL and UBA domains of UBQLNs are shown in figures.
  2. In Fig. 1A; It will be nice to illustrate that the accumulated p21 arrest G1-S transition by an arrow, since the current figure apparently seems to affects S-G2 transition.
  3. On pages 5-6; BCL2B10 should be BCL2L10.
  4. Please indicate UBQLN1 in Fig. 3, and insulin receptor is usually abbreviated as INSR on line 197.
  5. On line 264; please explain briefly about UFMylation, since it may not so be familiar with readers, unlike ubiquitination. Please consider to represent UFM in Fig. 5. Also, the title of Fig. 5 may be “UBQLN4 and DNA repair networking in cancer”, but not UBQLNs.
  6. On line 335, Figure 6C should be 6D, and the authors should cite Figure 6C somewhere between lines 318-325.

Author Response

Response to Reviewers #1

In this review, the authors summarized the pathophysiological functions of ubiquilin (UBQLN) family proteins on tumorigenesis. Characteristically, UBQLNs contain the N-terminal ubiquitin-like (UBL) and the C-terminal ubiquitin-associated (UBA) domains, and regulate multiple cellular functions. Importantly, genetic mutations in UBQLN are reportedly associated with amyotrophic lateral sclerosis (ALS). Therefore, the involvement of UBQLNs in neurodegenerative diseases have been extensively investigated hitherto. Intriguingly, this review sheds light on the role of UBQLNs on cancers by regulating prognosis, p53, cell death, membrane receptors, EMT, and miRNAs. Thus, this review will be helpful for all the readers who have interests on UBQLNs and tumorigenesis.

Point 1: On lines 31-35; Although the authors introduced the UBQLN1-4 and UBQLNL in this review, the existence of UBQLN5 and UBQLN6 are currently proposed in the Ref [7]. If it is possible, I hope the authors to represent the total residues and domain structures of human UBQLN family proteins as a figure, since the authors refer to the STI domains (on lines 148 and 198), whereas only UBL and UBA domains of UBQLNs are shown in figures. In Fig. 1A; It will be nice to illustrate that the accumulated p21 arrest G1-S transition by an arrow, since the current figure apparently seems to affects S-G2 transition.

Response to Point 1: We thank the reviewer for the suggestions. We agreed to add a new figure to show the total residues and domain structures of the major human UBQLNs that are discussed in this review and have been linked to human diseases.

In the present review, we have not included the UBQLNL, UBQLN5, and UQBLN6 protein domains because they have not so far been functionally linked to any human disease. Moreover, those UBQLNs have been poorly characterized so that limited information is available on their domain structures and functions.

The new Figure 1 and its caption are shown below,

Figure 1 Structural organization of human UBQLNs. 

The figure depicts the functional domains of major UBQLNs (UBQLN1-4) that have been functionally linked to human diseases, so far. Structural analyses of the UBQLN proteins were performed using the integrated tool InterProScan (available online at https://www.ebi.ac.uk/interpro/). UBQLN1, UBQLN2 and UBQLN4 proteins have very similar UBA and UBL domains, encode structurally identical proteins containing four STI domains and are related by successive duplications [7]. In contrast, UBQLN3 carries a single STI domain and has been proposed to be more evolutionary divergent. UBL: Ubiquitin-like domain; STI: STress Inducible proteins or Hsp70–Hsp90 organizing protein; UBA: Ubiquitin associating domain.   

The reviewer suggested that we illustrate that the accumulated p21 arrest G1-S transition by an arrow, since the current figure (the old figure 1A) apparently seems to affects S-G2 transition. We totally agreed and have modified the figure as suggested and this figure now is figure 2A (shown in the manuscript).

Point 2: On pages 5-6; BCL2B10 should be BCL2L10.

Response to Point 2: We apologize for our mistakes. We have changed all the BCL2B10 to BCL2L10 which is indicated in page 7-8, section UBQLNs and BCL2 family proteins. The fixed ones are shown below.

“UBQLN1 has been found to specifically interact and stabilize BCLb/BCL2L10, one of the six members of anti-apoptotic BCL2 family, via the first two of STI domains in the middle of UBQLN1 [44]. Normally, BCLb/BCL2L10 is localized in the mitochondrial and ER membranes and it is degraded in the cytoplasm by UPS [45–47]. Upon stimulation of UBQLN1 expression, BCLb/BCL2L10 is relocated into the cytoplasm, ubiquitinated on multiple lysine residues, and stabilized, increasing a chance of tumorigenesis and progression [46]. Despite a controversial role of BCLb/BCL2L10 in apoptotic-related functions, BCLb/BCL2L10 protein expression is inversely correlated with survival outcome of patients in various cancer types [46,48,49]. BCLb binds to Bax and suppresses apoptosis induced by Bax overexpression [50]. However, further studies to define its interactive partners in each of cancer-specific condition are required. For example, the ubiquitinated and non-ubiquitinated forms of BCLb/BCL2L10 can be stabilized by UBQLN1, and BCLb/BCL2L10-UBQLN1 interaction leads to an increase in oncogenicity in leukemic model [46] (Figure 2). Consistently, patients with primary lung adenocarcinomas are reported to have highly expressed UBQLN1 mRNA and protein in their tumors more than in an adjacent lung tissue [17].

            In addition, the binding of BCLb/BCL2L10 and BECN1 (Beclin 1), a central regulator of autophagy, also leads to an inhibition of autophagosome formation by disrupting BECN1-VPS34 complexes. Therefore, low expression of BCLb/BCL2L10 in hepatic cell carcinoma (HCC) tissues and cells can release free BECN1 to regulate autophagosome formation and lead to cancer cell survival [51,52] (Figure 3).”

AND another in New Figure 3 caption,

Figure 2 UBQLNs and BCLb/BCL2L10 networking in cancers

Point 3: Please indicate UBQLN1 in Fig. 3, and insulin receptor is usually abbreviated as INSR on line 197.

Response to Point 3: We thank the reviewer for the suggestion. We have indicated UBQLN1 in New Figure 4 and changed an abbreviation of insulin receptor from “INR” to “INSR” on line 263, section UBQLNs and membrane receptor proteins. Three revised points are shown below,

AND

“Moreover, this phenomenon has also been seen in the case of insulin receptor (INSR) and insulin-like growth factor 2 receptor (IGF2R) [64].”

Point 4: On line 264; please explain briefly about UFMylation, since it may not so be familiar with readers, unlike ubiquitination. Please consider to represent UFM in Fig. 5. Also, the title of Fig. 5 may be “UBQLN4 and DNA repair networking in cancer”, but not UBQLNs.

Response to Point 4: We thank the reviewer for the suggestion. We have added a short sentence describing about UFMylation. Moreover, the MRE11 UFMylation by UFM1 has also been indicated in New Figure 6 as well as the changing of New Figure 6 into “UBQLN4 and DNA repair networking in cancer”. Three revised points are shown below,

“Several studies provide evidence that ATM, UFMylated MRE11 (a conjugation form of MRE11 by ubiquitin-fold modifier 1 (UFM1)), and MRN complexes are considered to be specific factors processing toward HRR pathway [94].”

AND

AND

“Figure 6 UBQLN4 and DNA repair networking in cancer”

Point 5: On line 335, Figure 6C should be 6D, and the authors should cite Figure 6C somewhere between lines 318-325.

Response to Point 5: We thank the reviewer for the suggestion and apologize for our mistakes. We have changed the statement describing Figure 6C to New Figure 7D and also cited New Figure 7C on line 423. The changes are shown below.

“The miR-675 coding sequence lies within the first exon of the long-non-coding RNA (lncRNA) H19 [105]. miR-675 has been proven to cause reduction of cell proliferation, cell growth arrest, and cell death and thus acts as tumor suppressor in  NSCLC and other cancer cells [106–108]. Interestingly, Wang and co-workers showed that miR-675 controls the abundance of UBQLN1 in a fashion that augmentation of miR-675 leads to an increase in level of UBQLN1 which in turn reduces the level of ZEB1. Given the fundamental role of ZEB1 as a prime element of a network of transcription factors that controls EMT, the miR-675/UBQLN1 axis was shown to regulate the progression and development of pancreatic cancer by the regulation of ZEB1-dependent EMT [108] (Figure 7C).”

AND

“The majority of studies, revealing the ncRNA/UBQLN axis in cancer, focus on the effects of miRNAs on UBQLN1. Recently, new findings suggested that such a fine cell regulation also involves other UBQLNs, like UBQLN4. Indeed, Yu and co-workers showed that miR-370/UBQLN4 axis regulates the formation and progression of hepatocellular carcinoma [109]. Given the emerging role of UBLQN4 to regulate cell proliferation and invasion [29,110], the online tools TargetScan, miRTarBase, and miRcode were run out to further investigate the upstream regulators of UBQLN4. In vivo studies showed that miR-370 binds to the UBQLN4 3’-UTR and leads to its degradation. Interestingly, the TCGA database analysis also confirms that the expression of UBQLN4 negatively correlates with the miR-370 expression [109]. As such, miR-370 seems to modulate the Wnt-β-catenin pathway that is controlled by UBQLN4 and is responsible of HCC progression (Figure 7D).”

Reviewer 2 Report

The manuscript entitled "The Ubiquilin networking in cancers" by Jantrapirom et al. is a very timely and very well organized review dealing the the role of ubiquilins in cancers. It is a pleasure to read the manuscript, and the figures nicely illustrate the main statements written in the main text. Therefore, I recommend publication of this article in Cancers.

Minor points:

Please double check you manuscript. There are a few spelling and grammar errors.

Author Response

Thank you the reviewer very much, we have corrected and made some changes in the manuscript.